# ARIP: A Tool for Precise Interatomic Contact Area and Volume Calculation in Proteins

**DOI:** 10.3390/ijms25105176

**Published:** 2024-05-09

**Authors:** Tao Ma, Wenhui Li, Zhiping Tang, Xiangwei Sun, Lijuan Li, Zhonghua Liu, Gaihua Zhang

**Affiliations:** 1The National and Local Joint Engineering Laboratory of Animal Peptide Drug Development, College of Life Sciences, Hunan Normal University, Changsha 410081, China; xiangtao312@outlook.com (T.M.); whliabcc@gmail.com (W.L.); pingguotang498@gmail.com (Z.T.); xiangsun023@gmail.com (X.S.); ligs0116@gmail.com (L.L.); 2Peptide and Small Molecule Drug R&D Platform, Furong Laboratory, Hunan Normal University, Changsha 410081, China; 3Institute of Interdisciplinary Studies, Hunan Normal University, Changsha 410081, China

**Keywords:** residue interaction, interatomic contact volume, atomic overlap weighted algorithm, protein structure, quantitative analysis, molecular dynamic capture

## Abstract

The interplay patterns of amino acid residues are pivotal in determining the tertiary structure and flexibility of proteins, which in turn are intricately linked to their functionality and interactions with other molecules. Here, we introduce ARIP, a novel tool designed to identify contact residues within proteins. ARIP employs a modified version of the dr_sasa algorithm and an atomic overlap weighted algorithm to directly calculate the contact area and volume between atoms based on their van der Waals radius. It also allows for the selection of solvent radii, recognizing that not every atom in proteins can interact with water molecules. The solvent parameters were derived from the analysis of approximately 5000 protein and nucleic acid structures with water molecules determined using X-ray crystallography. One advantage of the modified algorithm is its capability to analyze multiple models within a single PDB file, making it suitable for molecular dynamic capture. The contact volume is symmetrically distributed between the interacting atoms, providing more informative results than contact area for the analysis of intra- and intermolecular interactions and the development of scoring functions. Furthermore, ARIP has been applied to four distinct cases: capturing key residue–residue contacts in NMR structures of P4HB, protein–drug binding of CYP17A1, protein–DNA binding of SPI1, and molecular dynamic simulations of BRD4.

## 1. Introduction

Proteins are essential components of life, playing pivotal roles in a myriad of biological processes, encompassing enzymatic reactions, immune responses, and signal transduction [1]. The three-dimensional structure of a protein is a direct reflection of its function, and the network of interacting residues within the protein governs its folding, dynamics, and interactions with other molecules [2]. Therefore, accurate analysis of these interactions is vital for comprehending protein function. The network of residue contacts and their dynamics provide a valuable perspective for elucidating the molecular mechanisms of proteins.

Currently, numerous computational tools have been developed to predict residue contacts, significantly contributing to protein structure prediction, e.g., AlphaFold2 [3]. Despite these advancements, the gold standard for defining residue contacts remains ambiguous and arbitrary. For instance, two amino acids are often considered to be in contact if the center distance between their Cβ atoms (or Cα for Glycine) is less than or equal to 8 Å [4], or if the nearest spheres of two atoms are less than 2 Å apart [5]. This ambiguity underscores the ongoing need for more precise and universally accepted criteria for defining residue contacts in proteins.

Various computational tools have been developed to calculate residue contacts in proteins, including CSU&LPC [5] and dr_sasa [6]. These tools primarily focus on the calculation of contact areas by considering the van der Waals (vdW) radius of an atom or the solvent molecule. They determine contact areas between paired atoms and the solvent-accessible surface area of each atom [7,8]. In these definitions, each atom is treated as a sphere with a radius equal to the sum of its own vdW radius and an additional distance as solvent radii to account for potential overlap between spheres. The contact area is defined as the surface area of one sphere that lies within another [6]. However, it is important to note that the contact area between different atoms is not equal, and the contact area alone may not fully capture the complexity of interatomic interactions.

In this context, we introduce ARIP (**A**nalysis of **R**esidues **I**nteraction in **P**rotein), a novel software tool designed to calculate the contact area and volume between contacting atoms directly. The contact volume is defined as the volume of one sphere encompassed by another, with the sphere representing an atom. The contact area is related to Coulomb force, which is inversely proportional to the square of the distance. The contact volume, on the other hand, may be related to binding energy, as it provides a common space for electron movement, making it a more informative parameter for developing scoring functions. ARIP goes beyond traditional methods that solely focus on contact area, recognizing the importance of contact volume in describing molecular interactions. By computing both surface area and volume, ARIP provides a more complete understanding of molecular structure and function.

ARIP is capable of analyzing multiple models within the same Protein Data Bank (PDB) file for molecular dynamic capture and multiple PDBs in the same directory for batch analysis [1]. This capability makes it a versatile tool for studying protein dynamics and interactions.

Water molecules are pivotal in mediating the interactions between amino acid residues. Conventional software tools, such as dr_sasa, have typically employed a uniform solvent radius of 1.4 Å for all atoms [6]. However, not all atoms within proteins are capable of interacting with water molecules. In this study, we analyzed approximately 5000 protein and nucleic acid structures containing water molecules, as determined through X-ray crystallography at resolutions better than 2 Å, to explore the interactions between water molecules and protein atoms. Our analysis confirmed the appropriate solvent radius for each type of atom. Our findings suggest that every oxygen atom in proteins is highly hydrophilic, and water molecules preferentially interact with oxygen atoms that have the largest contact area, potentially causing steric hindrance that hinders interactions between water molecules and neighboring atoms of the oxygen atom.

In the context of ARIP, the inclusion of the parameter “-p” is critical; it applies a fixed solvent radius of 1.4 Å specifically to hydrophilic atoms such as oxygen (“O”), nitrogen (“N”), and sulfur (“S”). This approach ensures that the solvent radius is tailored to the hydrophilic nature of these atoms, providing a more accurate representation of their interactions with water molecules.

By incorporating new parameters for solvent radius, ARIP employs an atomic overlap weighted algorithm for volume calculation, providing a more precise representation of interatomic interactions. The software has been used to analyze around 1151 protein structures obtained through Nuclear Magnetic Resonance (NMR) technology, aiding in the capture of dynamic residue contacts. It was specifically employed for key residue capture in NMR PDB files (e.g., 1MEK) [9], molecular dynamic simulation capture of the BRD4 protein (four states) [10], ligand binding capture of CYP17A1 structures with drugs [11], and DNA binding interactions of the SPI1 protein–DNA complex structures [12]. In summary, ARIP is a comprehensive and user-friendly software tool that supports the analysis of nucleic acids and other molecules stored in PDB format. As an open-source tool, it provides researchers with unrestricted access and usability, facilitating the study of protein structure, function, and interactions.

## 2. Results

ARIP, developed in Python 3, has been uploaded to GitHub at the address provided (https://github.com/YUYE-RainLeaf/ARIP/ (accessed on 6 May 2024). It is a cross-platform structure analysis tool that can be installed automatically online. The workflow of ARIP, as depicted in Figure 1, is organized into five branches. The core of the software relies on two algorithms for contact volume calculation: a modified version of the dr_sasa algorithm and the atomic overlap weighted algorithm.

### 2.1. The Interactions between Water Molecules and Proteins

Water molecules play a crucial role in facilitating interactions between amino acid residues within proteins. Software tools traditionally used for calculating residue–residue contacts have employed the solvated sphere radius of an atom in a protein. This radius is the sum of the radius of a water molecule (approximately 1.4 Å, considering the radius of oxygen since the radius of hydrogen is too small) and the atom’s own radius [6,13]. However, not all atoms in proteins are hydrophilic, and the use of a fixed solvent radius may not accurately reflect the diverse interactions present in protein structures.

X-ray crystallography often captures only a fraction of the water molecules present within protein structures. To define consistent distribution principles, it is essential to examine a vast array of protein structures that include water molecules. In our study, we analyzed around 5000 protein and nucleic acid structures, each with resolved water molecules, using X-ray crystallography with resolutions better than 2 Å. By calculating the spherical distances between protein atoms and the nearest water oxygen atoms (Figure 2, Figure 3, Appendix A), we were able to enhance our understanding of the hydrophilic interactions occurring within proteins.

Our analysis revealed a clear overlap between water molecules and hydrophilic protein atoms. The objective was to estimate the mosaic distances, which are indicative of the overlap between the van der Waals spheres of interacting atoms and are critical for determining the intensity and type of the interactions (Figure 2 and Figure 3). These mosaic distances are significant because they reflect the extent of overlap between the van der Waals spheres of atoms in contact, and they assist in elucidating the strength and character of the interactions. Moreover, this approach allowed us to consider the diverse hydrophilic properties of different protein atoms, thus providing a more precise depiction of the protein’s interaction network.

#### 2.1.1. Mosaic Distance and Threshold of the Mosaic Distance

The spherical distances separating protein atoms from their nearest water molecules were meticulously categorized by both amino acid type (AA) and atomic element (Category) using a bin size of 0.1 Å for numerical counting (Figure 2) and percentage determination (Figure 3). A pronounced overlap was identified between water molecules and the nitrogen and oxygen atoms of proteins (Figure 2a,b). The insights gleaned from Figure 2 and Figure 3 suggest that hydrophilic atoms can engage in robust interactions with water molecules, warranting the precise delineation of mosaic distances within the ARIP framework to facilitate accurate contact volume calculations. In the ARIP model, the maximum permissible mosaic distances for water molecules interacting with various hydrophilic atoms are defined as follows: 0.1 Å for nitrogen, 0.2 Å for oxygen, 0.3 Å for phosphorus, and 0.5 Å for sulfur atoms.

The ARIP algorithm is designed to incorporate water molecules, each with a diameter of 2.8 Å, as intermediaries in the interactions between pairs of atoms with hydrophilic atoms separated by up to 2.8 Å, while adhering to the specified thresholds for mosaic distances. This methodology allows ARIP to effectively model the role of water molecules in facilitating interactions between hydrophilic atoms within the intricate architecture of protein structures. These parameters come into play when utilizing the “-p” switch for the calculation of contact areas and volumes.

While water molecules possess the theoretical capability to mediate interactions between atoms separated by more than 2.8 Å through vibrational modes, the ARIP model treats them as static intermediaries in atomic interactions, with their diameter serving as the benchmark for confirming interaction eligibility. This simplification provides a uniform and measurable technique for evaluating the contributions of water molecules to the structure and function of proteins.

#### 2.1.2. Water Interactions and Steric Hindrance

Hydrophilic atoms within proteins form robust interactions with water molecules, yet the confined spaces within the protein structure often prevent neighboring or internal atoms from direct contact with water. Figure 3 illustrates the bin ratio distributions of the spherical distances between the nearest water molecules and the primary atom categories (C, N, and O) across the 20 standard amino acids. It is apparent that carbon atoms in proteins are predominantly hydrophobic, with an average spherical distance to water oxygen atoms of approximately 0.6 Å, as seen in Figure 3 and Appendix A. Interestingly, the carbon atom in glycine shows a relatively hydrophilic nature, and those in alanine and other charged amino acids also display a degree of hydrophilicity. Nitrogen atoms, excluding those in proline, are generally hydrophilic, while oxygen atoms are notably hydrophilic, especially in alanine and proline.

Figure 2a,b highlight two distinct peaks that represent the closest water molecules to the “N” and “O” atoms of the peptide plane. Figure 3 visualizes the layered arrangement of water molecules around the “C”, “N”, and “O” atoms of the peptide plane in proteins. The peak closest to zero indicates water molecules not influenced by other hydrophilic atoms, whereas the second peak represents those affected by interference. Water molecules are categorized based on the presence of such interference (Appendix A). In the non-interference group, the average spherical distance between carbon atoms (“C”) and water oxygen atoms is around 0.6 Å. However, for “N” and “O” atoms, this spherical distance shows significant variation. Notably, the average spherical distance between protein oxygen atoms (“O”) and water oxygen atoms is consistently the shortest (Appendix A). Approximately 62.47% of water molecules closest to oxygen atoms (“O”) in the non-interference group are unobstructed by other hydrophilic atoms, whereas only 33.32% of nitrogen atoms (“N”) are in such a position. This hydrophilic interaction can result in steric hindrance by the nearest water molecules, which may impede interactions with their adjacent atoms, particularly for the oxygen atoms of proteins.

Water molecules near interfering atoms contribute to the steric hindrance effects observed for the “N” and “O” atoms of the peptide plane, as indicated by the second peak, which is 2 Å from the first peak—too close for another water molecule. This hindrance highlights the complex interplay between water molecules and protein atoms within the dynamic protein structure.

In summary, the hydrophilic and hydrophobic properties of protein atoms play a pivotal role in their interactions with water, affecting protein function and stability. The varying spherical distances between water molecules and protein atoms, coupled with the layered water structure and steric hindrance, are crucial for comprehending the protein’s conformation and reactivity. The findings, as presented in Figure 2 and Figure 3 and supplemented by Appendix A, offer a comprehensive view of these interactions, essential for the study of protein structure and function.

### 2.2. ARIP Performance

Compared to dr_sasa, ARIP has many new features, as shown in Table 1. These features enable ARIP to be applicable in more scenarios.

ARIP processes a 32-residue, 232-atom PDB file on an x86 laptop (i7-1165G7 @ 4.7 GHz) in 25 s at default precision and 169 s at enhanced precision. To compare runtime, we selected 100 X-ray diffraction PDBs and 50 solution NMR PDB files from the Protein Data Bank [1] and analyzed them on an x86 laptop (i7-1165G7 @ 4.7 GHz). The IDs of these PDBs are shown in Appendix A.

During the analysis of NMR PDB files, which typically contain multiple models, we encountered frequent errors and interruptions when using dr_sasa. In contrast, ARIP can process each model individually, generating sub-files for each one. In this study, dr_sasa was exclusively utilized for analyzing X-ray PDB files, achieving an average runtime of 24.38 s. ARIP, on the other hand, exhibited an average runtime of 402.98 s in default precision mode, with detailed mode runtimes provided in Table 2.

It is important to note that the time required for area calculation was nearly identical for both software tools. However, ARIP demonstrated significantly slower performance in volume calculation. This difference in performance could be attributed to the additional complexity of the volume calculation algorithm used by ARIP, which requires more computational resources than the area calculation method.

### 2.3. Analyzing NMR Structures of Proteins with ARIP

The application of ARIP extends beyond static structural analysis, delving into the dynamic nature of protein structures as revealed by NMR spectroscopy. NMR, a powerful tool for elucidating protein dynamics, provides a multifaceted view of protein structures in solution, capturing the inherent flexibility and transient interactions that are often missed by static crystallography.

In our study, we harnessed the capabilities of ARIP to analyze 1151 protein structures determined by NMR, aiming to identify and quantify residue–residue contacts that are crucial for protein folding, stability, and function. One of these structures, PDB: 1MEK, was selected as an example to demonstrate the identification of critical residue–residue contacts within proteins. Then, our approach involved comparing contact areas and volumes within these structures using ARIP’s default parameters that account for the solvent radius (1.4 Å) for each atom, providing a more nuanced understanding of intra/inter-molecular interactions.

#### 2.3.1. Genuine Contact Verification: Threshold of Contact Area and Contact Volume

In this study, we delved into the intricacies of protein residue interactions by employing NMR data to establish thresholds that distinguish genuine contacts. Our aim was to navigate the complexity of protein interactions and identify biologically significant contacts within a dataset comprising 1151 protein structures. We used the ARIP tool to calculate contact areas and volumes (excluding the “-p” switch), integrating the data within a range of 0 to 5 Å^2^/Å^3^ for comparative analysis.

Our analysis revealed small peaks adjacent to zero for both contact area and volume in intra-chain interactions, suggesting a potential boundary for differentiating genuine contacts. We proposed thresholds of 0.5 Å^2^ for contact area and 0.25 Å^3^ for contact volume, which were derived from the observed peaks and are hypothesized to represent the lower limits of biologically relevant interactions. Notably, the sharper peaks for contact volume suggest that volume measurements may provide a more reliable indicator of credible contacts than area measurements.

The distribution of these peaks varied significantly between intra-chain and inter-chain interactions (Figure 4), underscoring the complexity of protein interaction networks. Heatmaps of contact areas and volumes revealed a substantial amount of contact noise in intra-chain interactions (Appendix A), suggesting that not all detected contacts may be functionally relevant. This observation underscores the importance of establishing thresholds to differentiate genuine interactions from a background of non-specific contacts.

Furthermore, the types of residue–residue contacts were expected to differ between intra-chain and inter-chain interactions due to variations in amino acid distributions within or on the protein surface (Appendix A). This insight is pivotal for understanding how proteins interact with their environment and other molecules, which is fundamental for their function.

#### 2.3.2. Capturing and Visualizing the Flexibility of PDI with Implications for Catalytic Function

The PDB entry 1MEK, which is associated with the gene P4HB encoding a precursor to protein disulfide isomerase (PDI), consists of 40 models that provide a comprehensive view of the protein’s conformational landscape. P4HB, a protein that is highly conserved across mammals, is known for its critical role in the folding process of other proteins through the catalytic management of disulfide bonds. The structural ensemble of 1MEK offers a unique opportunity to study the flexibility of PDI, which is essential for its catalytic function and is influenced by pH changes that can modulate the enzyme’s active site [14].

Appendix A presents the average residue–residue contact volume for 1MEK, where stable contacts, highlighted by red dots, are vital for maintaining the protein’s folded state. These contacts are crucial for the structural integrity of PDI, which is necessary for its function as a disulfide isomerase. In contrast, Appendix A illustrates the fluctuation in residue–residue contact volumes, indicating flexible regions within the protein that are essential for its functional dynamics [15,16,17]. This flexibility is particularly important around the active site, where pH-sensitive amino acid residues, such as those involved in the catalytic mechanism of PDI, can undergo conformational changes in response to changes in the cellular environment [14,18].

Two pairs of charged residues, A13R–A69E and A25K–A59E, are of particular interest due to their moderate average contact volumes (around 30 Å^3^) and a wide range (Max–Min) of contact volumes (exceeding 100 Å^3^). Using PyMOL [18] for visualization, the charge distribution for these residue pairs was analyzed (Figure 5), showing that the A13R–A69E pair had the smallest contact in Model 16, while the A25K–A59E pair had the largest contact in Model 29. The Root Mean Square Deviation (RMSD) between these models, as calculated using TM-align [19], was 1.52 Å for A13R–A69E and 1.34 Å for A25K–A59E. These residues, all located on the protein surface with high solvent accessibility, are likely to be key players in the conformational changes that PDI undergoes, which may be significantly influenced by solvent conditions and pH variations.

The pH dependence of PDI’s activity can be attributed to the ionization states of these critical residues, which can affect the enzyme’s redox potential and its ability to form and rearrange disulfide bonds. This understanding of PDI’s flexibility and pH sensitivity provides a foundation for further exploration of how PDI’s structure and function are intertwined, particularly in the context of its role in protein folding and the broader implications for cellular processes.

### 2.4. Abiraterone, Galeterone, and CYP17A1: Structural Insights into Drug Binding and Inhibition

The comparative analysis of abiraterone and galeterone (TOK-001) against human CYP17A1, a key enzyme in androgen biosynthesis, reveals significant differences in their inhibitory potencies. Abiraterone demonstrates a half-maximum inhibitory concentration (IC_50_) of 201 ± 1 nM, which is notably lower than that of galeterone at 503 ± 1 nM. This disparity in inhibitory strength is further elucidated by examining the structural interactions within the CYP17A1 enzyme, as depicted in the PDB entries 3RUK and 3SWZ [11].

A critical residue for the binding of both drugs is N202, which forms substantial interaction volumes exceeding 5 Å^3^. The interaction volume range (Max–Min) for N202 is considerably larger for galeterone (5.46 Å^3^) than for abiraterone (1.52 Å^3^), implying a less stable interaction and potentially contributing to the higher IC_50_ value observed for galeterone. Additionally, two other residues, D298 and Y201, known to be crucial for ligand binding, show interaction volumes greater than 1 Å^3^ for both drugs. However, the average interaction volumes for abiraterone are higher than those for galeterone (Appendix A), suggesting a more stable and potentially tighter binding mode for abiraterone.

These structural insights, derived from the crystallographic analysis, provide a molecular rationale for the observed differences in IC_50_ values between abiraterone and galeterone. The binding mode of these inhibitors is substantially different from those predicted by homology models or from steroids in other cytochrome P450 enzymes with known structures [11]. Interestingly, the interactions with C442 and the heme group (HEM600) were not as expected, indicating the need for molecular dynamic simulations to further dissect the dynamic nature of these interactions and their implications for CYP17A1 inhibition.

The structural information presented here is instrumental for understanding the functional implications of CYP17A1 in the context of prostate cancer therapy. It also facilitates the rational design of more selective and potent inhibitors, which could be pivotal in the development of targeted treatments for hormone-responsive cancers.

### 2.5. SPI1 and DNA Binding

The master transcriptional regulator PU.1, also known as Spi-1, exhibits a distinct preference for purines at the Q226 position, which is integral to its DNA binding specificity. This preference is pivotal in understanding the pathogenesis of the Q226E mutation observed in Waldenström macroglobulinemia, a finding that aligns with the high-resolution PU.1 structures presented by Terrell et al. [12]. Furthermore, our study suggests that K237 plays an essential role in SPI1–DNA binding, as its DNA-binding capability is compromised in the E226 mutation context. This observation is supported by the structural analysis of SPI1–DNA complexes, where K237’s interaction with DNA is significantly diminished, potentially affecting the overall DNA binding affinity.

Additionally, the R220 residue, which is also involved in DNA binding, shows a slight reduction in its interaction with DNA in the E226 mutation context. The SPI1–DNA contact map reveals that while most amino acid residues interact with one or two DNA bases, R230 stands out by strongly interacting with three consecutive base pairs. Interestingly, the interaction volumes between Q226 and E226 with DNA are comparable, indicating that the mutation may lead to weakened DNA binding by other basic amino acids, which could be a critical factor in pathogenesis.

Upon comparing the protein structures with the Q226 and the E226 mutation in SPI1, we observe an intriguing shift in residue–residue interactions. Specifically, the mutation from Q226 to E226 enhances the contact between R230 and R233, which could precipitate structural changes in the protein, as depicted in Figure 6b, Appendix A. Moreover, the Q226E mutation is linked to changes in the charge distributions of K237, R230, and R233, which may profoundly affect the protein’s DNA binding functionality. These altered charge distributions could disrupt the electrostatic interactions essential for stabilizing the protein–DNA complex, offering a plausible explanation for the reduced DNA binding affinity associated with the Q226E mutation.

Our findings, supported by high-resolution structural data from Terrell et al. [12], emphasize the significance of evolutionary innovations like Q226 in PU.1 and their role in preserving the structural and functional integrity of transcription factors. The comprehensive understanding of SPI1–DNA interactions, especially in the context of pathogenic mutations, forms a basis for further research into the molecular mechanisms driving transcriptional dysregulation in diseases like Waldenström macroglobulinemia.

### 2.6. Molecular Dynamic Simulation Capture of 5ULA: Unveiling Key Interactions in BRD4

The bromodomain-containing protein 4 (BRD4) is equipped with a bromodomain that interacts with acetylated marks on histones, a critical interaction for epigenetic regulation. To elucidate the transient states and dynamic behavior of BRD4, Lluís Raich and colleagues conducted extensive molecular dynamics simulations (MDS) [10]. Our analysis focused on the PDB structure 5ULA, scrutinizing three distinct states derived from the MDS data. A meticulous examination of the interaction volumes’ average and range (Max–Min) revealed key residue–residue contacts (D106–T109, D106–P86, and M107–Q84) that were previously reported to participate in conformational transitions, as detailed in Raich et al.’s study [10].

Additionally, our investigation uncovered another cluster of contacts (N121–E124, R58–W120, and R58–N117) from a distinct loop that may also exert a significant influence on the conformational transitions. The residue R58, similar to D106, is potentially sensitive to pH changes, which could have implications for the protein’s interaction with its environment and other cellular components (Appendix A). BRD4, a nuclear-localized protein, plays a pivotal role in gene expression through its interaction with acetylated histones. The alkaline nature of histones allows them to interact with DNA, which is acidic, underscoring the importance of pH-sensitive amino acids like R58 and D106 in modulating BRD4’s conformation. This molecular interaction aligns with the principles of physical chemistry, as the sensitivity of these residues to pH changes enables the fine-tuning of BRD4’s structural dynamics in response to cellular environments. These findings enhance our understanding of the dynamic nature of BRD4 and its interaction with the cellular milieu, providing new insights for the development of targeted therapies for diseases where BRD4 plays a critical role.

### 2.7. Formatting of Mathematical Components

The calculation of interacting contact volumes follows a method akin to dr_sasa (default), ensuring that overlapping volumes are not double counted. Since volume incorporates an additional dimension relative to area, the grids are dispersed throughout the entire space rather than being confined to the surface (as shown in Equation (1)).
(1)V=∆v×∑(1/ci)

Here, ***V*** represents the contact volume between two atoms, Δ***v*** is the volume represented by each grid point on the atom, Σ signifies the sum of all the grids involved in contact, and ***ci*** represents the count of atoms in contact with the given atom within a grid. If ***ci*** equals 2, it means there are three atoms in contact in the target grid.

ARIP also offers an alternative atomic overlap weighted algorithm for calculating contact volumes. With this algorithm, the volume represented by each grid is multiplied by the count of overlapping atoms (as shown in Equation (2)).
(2)$$V_{0}=∆v\times \sum \left(1+ci\right)$$

***V***_0_ represents the weighted volume between two atoms, with Δ***v*** being the volume represented by a grid point on the atom and *(***1**
*+ **ci**)* denoting the total number of atoms in contact with each grid, including the atom itself. This method is innovative and can be particularly useful when researchers deem regions with a higher degree of atomic overlap to be more significant.

When one atom makes contact with another, the contact surface area is determined by counting the number of grids on the surface of the first atom that lie within the second atom. Each of these grids is multiplied by the surface area it represents to calculate the total contact surface area. However, it is possible for a single grid on the first atom to be in contact with multiple atoms simultaneously. To prevent double counting, the surface area represented by that grid is divided by the number of atoms it is in contact with before being tallied (as shown in Equation (3)).
(3)S=∆s×∑(1/ci)

Here, ***S*** denotes the contact surface area between two atoms and Δ***s*** is the surface area represented by each grid on the atom. In ARIP’s calculations, a default of 5000 grids is assigned to each atom, with an option to use 15,092 grids for a higher precision mode.

The CSV file with the “_RES” prefix provides data on the contact surface area, volume, and AOWV for each pair of residues. These values are cumulative, reflecting the sum of the respective metrics for all atoms that interact between the two residues (as described by Equations (4) and (5)).
(4)$$S_{r}=\sum S_{a}$$
(5)$$V_{r}=\sum V_{a}$$

Here, ***S_r_/V_r_*** denotes the total contact surface area/volume between two residues, while ***S_a_/V_a_*** represents the contact surface area/volume for each individual pair of interacting atoms within those residues. By offering various output files, researchers can readily access the necessary information and further analyze it to elucidate the underlying biological significance.

## 3. Discussion

Proteins can be likened to intricate buildings constructed from a diverse set of twenty amino acids, with water molecules serving as the bricks that bind these components together. Within this architectural marvel, certain residue–residue contacts act as the main structural walls that dictate the protein’s folding pattern, akin to the load-bearing walls in a building. Conversely, other contacts function similarly to doors and windows, which is pivotal in determining the protein’s functionality and allowing for the dynamic exchange with its environment. Distinguishing between these contact types and dynamically capturing their interactions is fundamental when investigating the protein structure.

This study introduces the ARIP tool, a novel method for accurately calculating interatomic contact area and volume within and between proteins. Our application of ARIP has not only deepened our understanding of the complexity of protein structures but also paved new avenues for exploring protein functions. The results underscore the critical role of water molecules in mediating interactions between residues, offering profound insights into the solvation characteristics of proteins. Water molecules, as mediators in biomolecular reactions, are integral to protein folding, stabilization, and the execution of their functions. They form bridges that facilitate essential hydrogen bonds between residues, maintaining the protein’s three-dimensional structure. Moreover, water molecules at protein active sites significantly influence substrate binding and catalytic reactions. ARIP provides a platform to observe these effects, offering fresh perspectives on protein interactions within their hydration layer and external environment. Future research could delve into the specific roles of water molecules in protein–protein and protein–small molecule interactions, particularly in the realms of drug development and protein engineering.

The incorporation of contact volume calculation and the atomic overlap weighted algorithm into ARIP marks a significant methodological advancement in our study. This approach provides a more nuanced understanding of the interatomic interactions that are fundamental to the structure and function of proteins. The Coulomb force, a critical determinant in atomic interactions, diminishes with the square of the distance between charged particles. However, at the atomic scale, the interaction is more complex due to the distribution of electrons around the positively charged nuclei, which are not uniformly distributed and can lead to variable forces at different points within the atomic cloud.

Our modified dr_sasa algorithm and the atomic overlap weighted algorithm within ARIP are designed to account for these complexities by calculating the volume of space overlapped by interacting atoms, rather than just the surface area of contact. This volume, we posit, may correlate with the binding energy between atoms, as it represents the space available for electron delocalization—a key factor in chemical bonding. The atomic overlap weighted algorithm further refines this concept by adjusting the contact volume based on the degree of atomic overlap, suggesting that areas with greater overlap could be indicative of stronger or more significant interactions.

This hypothesis—that the total overlapped space between atoms could serve as a parameter for evaluating binding energy—opens a new avenue for understanding the stability and dynamics of molecular interactions. However, we acknowledge that the relationship between contact volume and binding energy is not yet fully understood and requires further investigation. The nature of the chemical bonds formed at the points of contact, whether covalent, ionic, or hydrogen bonds, is pivotal in determining the biological relevance of these interactions.

Our study with ARIP is a preliminary step towards characterizing these interactions in a quantitative manner. The results from our analysis provide a foundation for future research that could lead to a more sophisticated comprehension of the interplay between atomic contact volume and the physical properties of molecular binding. As we refine ARIP and our understanding of these interactions, we move closer to a deeper appreciation of the molecular mechanisms that govern protein function and stability.

ARIP has proven instrumental in analyzing the dynamic structures of proteins and protein–protein interactions, crucial for understanding biomolecular networks and signal transduction pathways. The study emphasizes the importance of accurately quantifying atomic contacts within proteins, which is vital for understanding how proteins achieve specific functions through their three-dimensional structures. This knowledge lays a valuable foundation for protein engineering and drug design. By precisely determining contact areas and volumes around protein active sites, we enhance our ability to predict small molecule drug binding affinity, aiding in the design and optimization of drug molecules. While ARIP shows promise in protein structure analysis, it is not without limitations, particularly its reliance on high-quality protein structure data. This dependency may limit the analysis results for proteins with low resolutions or partially unknown structures. Nevertheless, the tool’s potential impact on the study of protein interactions and the development of new therapies is significant.

## 4. Materials and Methods

### 4.1. Dataset

For the purposes of this study, we utilized three primary datasets. The first dataset consisted of roughly 5000 X-ray crystal structures with water molecules, each having a resolution of less than 2 Å, which were randomly chosen using a custom Perl script. The second dataset included 1151 protein structures determined by NMR spectroscopy (Appendix A) selected using our Perl script and de-duplicated with cd-hit version 4.8.1 to ensure structural diversity. The third dataset, used to evaluate the software’s processing speed, is provided in Appendix A.

Building upon these datasets, we further demonstrated the functional applications of ARIP using four additional examples. These included the analysis of the NMR structure of PDB 1MEK, the examination of protein–DNA interactions in 22 SPI1 complexes with DNA [12], the investigation of protein–ligand interactions in 2 CYP17A1 structures with bound ligands [11], and the identification of key residues during molecular dynamics simulations using 3 structures from MD simulations of PDB 5ULA [10]. These examples showcased the versatility of ARIP in various biological contexts, highlighting its utility in understanding protein structure and function.

### 4.2. Mosaic Distance, Layer of Solvent, and Steric Hindrance

In reality, it is not reasonable to assume that every protein atom is in contact with a water molecule, with the solvent layer defined as the van der Waals (vdW) radius of the atom plus 1.4 Å [20,21,22,23]. To address this issue, we analyzed the aforementioned X-ray structures to estimate the mosaic distance between water oxygen atoms and protein atoms. We employed a modified version of the ARIP tool to calculate the spherical distances between water molecules and protein or nucleic acid atoms, which are the center distances minus the sum of the two atoms’ radii. We then collected data on the number of nearest water molecules to each type of residue atom for distance distribution analysis, using a bin size of 0.1 Å (Figure 2). The spherical distance range analyzed extended from −2.0 to 2.8 Å.

In this study, we identified the nearest water molecules to the carbon (“C”), nitrogen (“N”), and oxygen (“O”) atoms within the peptide plane, distinguishing between those that are specifically and commonly associated with these atoms. We categorized these water molecules into two groups: one where the carbon, nitrogen, and oxygen atoms are also the nearest to these water molecules (without interference) and another where there are other atoms that are closer to the water molecules than the carbon, nitrogen, and oxygen atoms (with interference). We then calculated the percentage and spatial distribution of these atoms and their associated water molecules (HOH pairs) within their respective sets to validate our hypothesis regarding the solvent layer and the steric hindrance on water molecular interactions. This analysis provides insights into the complex interplay between protein atoms and water molecules, shedding light on the molecular mechanisms that underpin protein structure and function.

### 4.3. ARIP: Function and Principle

ARIP has extended the grid-based algorithm used in dr_sasa [6,13] to also enable the quantification of the volume of atoms in contact. ARIP employs the same method as dr_sasa for calculating contact area (see Section 4.3.3 for more details), where atoms are represented as spheres with a radius equal to their van der Waals (vdW) radius plus the radius of a water molecule (1.4 Å by default). The surface area and volume of these spheres are divided into grids (Appendix A). Each grid represents a small surface area or volume element, and the grids where contact occurs between two spheres are tallied to compute the total contact surface area and volume.

The ARIP workflow comprises five main branches (Figure 1).

#### 4.3.1. Contact Judgement from Branch A

In ARIP, the constants for atoms, such as atomic radii, are derived from dr_sasa. Residue–residue contacts are determined by assessing whether the center distance between any atom of one residue and any atom of the other residue is less than 2.8 Å.

The atom types and contact types in ARIP are based on the classifications used in CSU&LPC and dr_sasa. ARIP categorizes interactions in proteins into various types according to atom type and distance. For covalent interactions, the program specifically identifies peptide bonds and disulfide bonds. Disulfide bonds are recognized when the distance between two sulfur atoms on cysteine residues falls within the range of 1.95 Å to 2.1 Å [24]. Non-covalent interactions are categorized based on the combinations of eight types of heavy atoms found in standard protein residues, leading to categories such as Hydrogen Bond (HB), Aromatic–aromatic Contact (AROM), Hydrophobic–hydrophobic Contact (PHOB), Destabilizing Contact (DC), and other miscellaneous non-covalent interactions (OTHER) [5]. HB interactions are defined as contacts between hydrogen bond donor and acceptor atoms within a distance of 1.5 Å to 3.5 Å [25]. AROM interactions involve contacts between atoms on the aromatic ring of aromatic amino acids, and the three carbon atoms CG, CD2, and CE1 in the imidazole group of histidine are treated as aromatic atoms [5]. PHOB interactions arise from contact between hydrophobic atoms, while DC interactions occur between hydrophobic and hydrophilic atoms.

Building on the classification of contact types between atom pairs, interactions can also be categorized into Short (S), Middle (M), Long (L), and Inter-chain (I) interactions based on the distance between residues in the protein’s primary structure. Short-range interactions occur between neighboring residues, middle-range interactions span several residues, long-range interactions cover more distant residues, and inter-chain interactions involve residues from different protein chains (Table 3) [4,10,11,12]. A high number of short interactions may significantly contribute to the stability of protein structures, while long interactions could be associated with changes in protein conformation.

#### 4.3.2. Contact Volumes Calculation from Branch B

The fundamental operation of ARIP revolves around calculating contact volume, for which it offers two distinct levels of granularity: basic and enhanced. The basic level employs a cubic lattice with a spacing of 0.2 Å to determine the volume, while the enhanced level level ups the resolution to 0.1 Å intervals for a more detailed computation.

ARIP incorporates two algorithms and two methods for defining solvent radii in its contact volume calculations. One algorithm is an adaptation of dr_sasa, and the other is a novel atomic overlap weighted method. The first method of defining solvent radii is consistent with dr_sasa, where each atom is treated as a sphere with a radius equivalent to the sum of its van der Waals (vdW) radius and the radius of a water molecule (1.4 Å by default). The alternative definition of solvent radii is founded on an examination of crystal structures containing water molecules, focusing on interactions where at least one atom is hydrophilic, such as oxygen or nitrogen atoms.

##### Hydrophilic Atom Interactions Mediated by Water Molecule

ARIP employs a filtering mechanism to exclude atom pairs that do not include at least one hydrophilic atom (N, O, P, or S). This step ensures that only atom pairs with the potential for water-mediated interactions are considered. To establish the presence of such interactions, the closest distance between the spheres representing the atoms must be less than 2.8 Å. In this process, a water molecule is virtually positioned at a distance that corresponds to the mosaic distance of the hydrophilic atom.

For atom pairs in which only one atom is hydrophilic, the calculated volume reflects the contact this atom can have with a single water molecule. In cases where both atoms in the pair are hydrophilic, the calculated volume is an average of the contact each atom can have with a water molecule, taking into account that there is no overlap between the atomic spheres (refer to Figure 2f for a visualization of this process).

##### Calculation of the Ordinary Contact Volumes

To calculate the interacting contact volumes, ARIP’s default method follows the approach used for calculating contact surface areas in dr_sasa. However, it expands on this by populating grids throughout the entire space, rather than just the surface shell. This comprehensive coverage allows for the inclusion of the third dimension, thereby providing a more accurate representation of the volume being interacted with (Appendix A). This method ensures that the complex spatial relationships within the volume calculation are effectively captured (refer to Equation (1) for the default calculation).

One important aspect of this algorithm is that it avoids double counting the contacted volume. This means that once a volume element has been identified as part of an interaction, it is not counted again in subsequent calculations, ensuring that the final volume measurement is an accurate reflection of the unique spatial extent of the interaction.

##### Calculation of the Atomic Overlap Weighted Volumes (AOWV)

ARIP provides an alternative method known as the atomic overlap weighted algorithm for calculating contact volume. This approach introduces a refinement by multiplying the volume of each grid point by the number of atoms that overlap it. This step explicitly considers the cumulative effect of atomic overlaps on the overall volume calculation (refer to Equation (2) for the specific calculation).

The use of this algorithm allows the contacted volume to serve as a proxy for the electron density, which in turn can be related to the binding energy of the interaction or the difficulty of dissociation for the atoms involved. By incorporating the overlap factor, the atomic overlap weighted algorithm provides a more nuanced understanding of the molecular interactions, particularly in cases where atomic proximity and overlap significantly influence the stability or energetics of the system.

#### 4.3.3. Contact Area from Branch C

ARIP employs two methods for defining solvent radii in its contact area calculations, which are analogous to those used for volume calculation. The default method uses the same algorithm and solvent radii definition (1.4 Å) as dr_sasa. The alternative method involves filtering out atom pairs that do not include at least one hydrophilic atom (N, O, P, or S), with the spheres’ closest distance being less than 2.8 Å to ensure the presence of water-mediated interactions. For atom pairs with only one hydrophilic atom, the calculated area represents its contact with one water molecule; for pairs where both atoms are hydrophilic, the values are the average of each atom’s contact with one water molecule, without considering atomic overlaps (refer to Figure 2f).

ARIP offers two levels of precision for calculating contact area: default and enhanced. The default precision uses a spherical grid consisting of 5000 points to calculate the surface area. For enhanced precision, ARIP utilizes a grid with 15,092 points, which provides a more detailed surface area calculation.

In ARIP, both the quantification of surface area and volume rely on grid approximation. Increasing the number of grid points enhances the accuracy of the calculations. Nevertheless, for large-scale data analysis, the default precision is sufficient for efficient calculations without compromising too much accuracy.

#### 4.3.4. Data Merge and Output from Branch D

ARIP is a powerful command line tool designed for the analysis of PDB files or directories containing multiple PDB files, enabling batch processing for efficient analysis [1]. The tool provides a range of options to tailor the analysis, including specifying output directories (-o), activating enhanced precision mode (-e), defining interactions mediated by water molecules between hydrophilic atoms (-p), setting a custom distance for atom contacts (-d), calculating solvent accessible surface area (SASA) (-a), employing an atomic overlap weighted algorithm (-w), saving results in a compressed format (-z), and specifying the number of threads to use for parallel processing (-t).

The output generated by ARIP includes CSV files with detailed information on contacting atoms, a summary of residue interactions, and dihedral angles. It can organize outputs into separate folders for each input PDB file and create subfolders for NMR files with multiple models. Optionally, it can produce CSV files for each residue (-r) and exclude contacts below a specified threshold (-c). ARIP also has the capability to skip volume calculation and focus solely on surface area and other relevant data (-s).

When processing PDB models, ARIP outputs atomic contact information to CSV files. The file with the _ALL prefix contains columns for Residue1, Atom1, Type1, Residue2, Atom2, and Type2, which describe the interacting pairs of residues and atoms. The AOWV (Atomic Overlap Weighted Volume) column is included when the -w switch is used. If the -s switch is employed, the Volume column is excluded from the output.

An A _SUM prefixed CSV file is produced for each model, providing a summary of residue-level information. Non-standard residue names are appended with “_”. Phi and Psi angles are shown only for standard amino acids. Covalent (Cova) and non-covalent (NC) contacts are indicated. The SASA, Cova_AOWV, and NC_AOWV columns are included when the “-a” or “-w” switches are activated. When the “-s” switch is used, the Cova_Volu and NC_Volu columns are omitted. UNDEF columns are included for non-standard residues with ambiguous link types.

The _RES prefixed CSV file contains the contact surface area, volume, and AOWV values for each pair of residues, representing the cumulative sum of these values for all contacting atoms between the two residues (refer to Equations (4) and (5) for the calculations). This detailed breakdown allows for a comprehensive analysis of the molecular interactions within the protein structures.

#### 4.3.5. Dynamic Capture and Visualization from Branch E

ARIP recognizes the inherent flexibility of residue–residue contacts, which is a crucial aspect of protein dynamics. To address this, ARIP offers a functionality that generates residue contact heatmaps for structures with multiple models. This feature allows researchers to identify significant residue pairs based on variations in contact volume or frequency, providing insights into the dynamic interactions within proteins. This model is particularly valuable in molecular dynamics simulations and for analyzing NMR structures, as it enables the creation of contact map images and visual representations of the range of contact volumes from the output matrices (see Section 4.3.4 for further details).

In the context of a specific study, the PDB structure 1MEK was used as an example to demonstrate this functionality (excluding the “-p” switch). Appendix A in the study illustrate the average and range of contact volumes between residues, respectively. In Appendix A, red dots represent flexible contacts (as shown in Figure 5) that are likely to be important for protein function [15,16,17]. These flexible contacts can undergo conformational changes that are critical for the protein’s biological activity.

Moreover, ARIP recognizes the importance of stable distant residue–residue contacts in maintaining protein stability. These long-range interactions contribute to the overall fold and structural integrity of proteins.

In addition to its analytical capabilities, ARIP also provides a tool for visualizing the spatial positions of heavy atoms in PDB structures (Appendix A). This feature is useful for examining the three-dimensional arrangement of atoms and can generate images for each model in PDB files that contain multiple models. This visualization tool complements the quantitative analysis offered by ARIP, providing a comprehensive understanding of protein structure and dynamics.

### 4.4. ARIP and Protein Structures from NMR

ARIP is tailored to process PDB files that may contain multiple models, providing a structured output that reflects this. For PDB files with only one model, ARIP generates a single output folder. However, for files with multiple models, such as those derived from NMR spectroscopy, ARIP creates corresponding subfolders for each model’s output. This organizational structure facilitates the analysis of complex protein dynamics.

In the study, 1151 protein structures determined by NMR were analyzed as a case using the default parameters (excluding the “-p” switch). The analysis yields a comprehensive set of data, including a summary CSV file with the “_ALL” prefix that details atomic contact information for each residue. A “_SUM” prefixed CSV file provides amino acid-level results, incorporating data such as dihedral angles (Phi and Psi), surface area, volume of each residue involved in covalent or non-covalent contacts, and solvent-accessible surface area [26,27,28,29,30,31].

ARIP also offers the capability to generate individual CSV files for each residue, allowing for a more focused examination of specific interactions. Users can opt to exclude volume calculations and solely compute surface area by adjusting the software settings, which can be useful for specific research questions or when computational efficiency is a priority (see Section 4.3.4 for further details).

Due to variations in the vdW radii of different atoms, the curvature of spheres will differ, leading to discrepancies in surface area calculations. When a larger sphere overlaps with a smaller one, the surface area within the overlap region will differ, while the enclosed volume remains constant. This underscores the advantages of volume measurements over surface area calculations in terms of accuracy. However, the use of default parameters, such as increasing the solvent radii of hydrophobic atoms by 1.4 Å, can introduce minor contacts that may be artifacts of the spherical approximation (Figure 4). To address this, ARIP permits users to set custom lower cutoff values for surface area and volume calculations, allowing for the exclusion of such artifacts and ensuring the integrity of the analysis.

### 4.5. ARIP and Protein–Ligand Interactions

ARIP is capable of identifying key residues involved in ligand binding, which is crucial for understanding the molecular basis of protein–small molecule interactions. In an illustrative analysis, two structures of the cytochrome P450 enzyme CYP17A1 complexed with different ligands [11] were examined using ARIP. The PDB identifiers for these structures are 3RUK and 3SWZ. Each PDB file in this study contains four models, with the first model in 3RUK featuring the steroidal inhibitor abiraterone (chemical formula C_24_H_31_NO), and the corresponding model in 3SWZ containing galeterone (also known as TOK-001, with the chemical formula C_26_H_32_N_2_O).

To investigate the interactions between the protein residues and these ligands, ARIP was employed with the “-p” switch to calculate the interaction volumes. This calculation provides a quantitative measure of the extent of binding between the protein and the ligand. Heatmaps were utilized to visualize the outcomes of these calculations (refer to Appendix A), providing a visual depiction of the interaction volumes. Heatmaps are an effective way to visually identify residues that have significant interactions with the ligands, which can be pivotal in drug design and understanding the mechanisms of ligand action.

### 4.6. ARIP and Protein–DNA Interactions

ARIP’s versatility extends to the analysis of DNA–protein interactions, which are fundamental to a wide range of biological processes, including gene regulation. In a particular study, the DNA sequences from 22 SPI1–DNA complexes were aligned to facilitate structural selection and group classification (see Appendix A). This alignment step is essential for understanding the conserved and variable aspects of the DNA binding sites and for categorizing the complexes based on their structural features.

Following the sequence alignment, 16 protein structures with DNA binding were chosen for a detailed case study [12]. These structures were organized into four groups based on the observed variations in the protein SPI1 and the DNA sequences with which it interacts (for specific details, refer to Appendix A). This classification allows for the comparison of DNA–protein interactions across different structural contexts.

ARIP was then employed with the “-p” switch to calculate the interaction volumes between the protein residues and the DNA bases. By determining the average interaction volumes, the study aimed to quantify the extent of binding between specific residues and bases, providing insights into the strength and specificity of these interactions (see Figure 6 and Appendix A). This analysis can reveal key residues that are pivotal for the stability and function of DNA–protein complexes, contributing to a deeper understanding of gene regulatory mechanisms and potentially informing the design of targeted interventions.

### 4.7. ARIP and Protein Molecular Dynamic Simulation

Then, ARIP (with “-p”) was utilized and the average and the Max–Mins of interaction volume value of each residue–residue contact were calculated.

ARIP is a valuable tool for identifying key residues in protein structures derived from molecular dynamics simulations, which are essential for understanding protein dynamics and ligand interactions. In a specific analysis, four protein models from molecular dynamics simulations of the bromodomain protein BRD4 (PDB identifier: 5ULA) were selected from the research conducted by Lluís Raich and colleagues [12]. These models represent three distinct states from the Dynamic Structural Modelling (DSM) approach.

To investigate the interactions between residues in these protein models, ARIP was employed with the “-p” switch to calculate the interaction volumes. This calculation provides a quantitative measure of the extent of binding between residues over the course of the simulation. The average and the range (Max–Min) values of the interaction volume for each residue–residue contact were determined (Appendix A). This analysis allows for the identification of residues that exhibit significant changes in interaction volume, which may be indicative of flexible or dynamically important regions within the protein. Such residues are likely to play critical roles in protein function and ligand binding, and they can serve as targets for drug design or further experimental investigation.

## 5. Conclusions

ARIP stands out as a robust and flexible structure analysis tool that is compatible with multiple operating systems. It provides researchers with a comprehensive set of parameters for calculating residue–residue contact areas and volumes, allowing for the customization of analyses based on various hypotheses about protein dynamics. The tool’s versatility makes it an invaluable asset for studying the intricate movements and interactions within protein structures.

Looking forward, our goal is committed to enhancing the tool’s capabilities, aiming to improve its performance and accuracy in handling complex protein structures and large-scale protein datasets. These enhancements will involve refining the algorithms used for contact area and volume calculations, as well as optimizing the tool’s computational efficiency to handle big data.

Additionally, the integration of bioinformatics and experimental data will be a priority, as this will enrich the tool’s ability to provide insights into protein functions and protein–protein interaction networks. This multidisciplinary approach will ensure that ARIP remains at the cutting edge of structural biology research and will provide a strong foundation for advancements in biomedical research and drug discovery.

In conclusion, ARIP is a powerful tool that is not only contributing to our understanding of protein dynamics but is also positioning itself as a key player in the field of structural biology, with the potential to significantly impact the study of protein interactions and the development of new therapies.

## Figures and Tables

**Figure 1 ijms-25-05176-f001:**
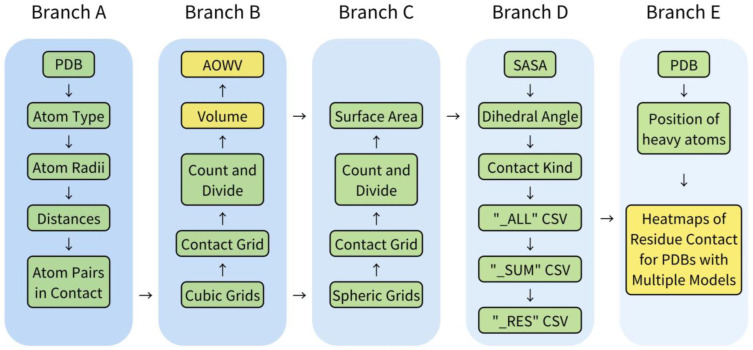
Workflow of ARIP. Branch A, constants of all types of atoms, and judgment of contacts; (ii) Branch B, contact volume calculation; (iii) Branch C, contact area calculation; (iv) Branch D, data merging and output; and (v) Branch E, visualization section.

**Figure 2 ijms-25-05176-f002:**
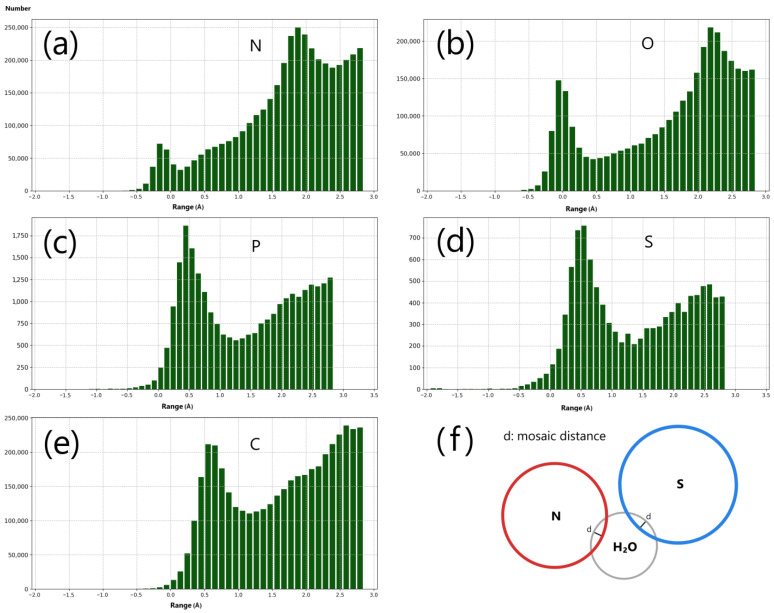
Spherical and Mosaic distances. (**a**–**d**) Distribution of spherical distances from hydrophilic atoms (N, O, P, and S) to their nearest water molecules, where negative values (spherical distance < 0) indicate mosaic distances; (**e**) distribution of spherical distances between carbon atoms and their nearest water molecules (control); (**f**) illustration of mosaic distances, using nitrogen and sulfur atoms as examples. The spherical distance between N and S is less than 2.8 Å and the inserted water molecule aims to maintain the mosaic distance (**d**) as much as possible in the ARIP framework.

**Figure 3 ijms-25-05176-f003:**
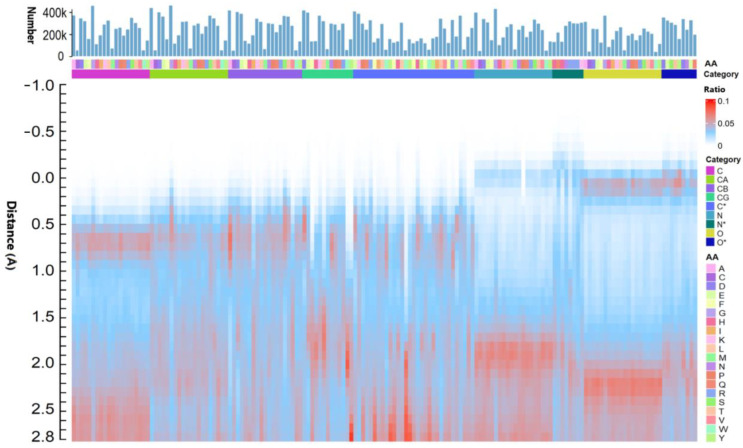
The spherical distances between carbon ©, nitrogen (N), and oxygen (O) atoms in 20 amino acid residues and their nearest water molecules exhibit a distribution ranging from −1 Å to 2.8 Å. C* includes carbon atoms such as CD, CE, CZ, CH, and other carbon atoms in amino acids; N* encompasses nitrogen atoms other than N in the peptide plane; O* comprises oxygen atoms other than O in the peptide plane. The spherical distance bin size is 0.1 Å. The ratio indicates the number of interactions in each bin divided by the total number of interactions within the range of −1 Å to 2.8 Å. The number of interactions between protein atoms and their closest water molecules is presented as a bar graph at the top.

**Figure 4 ijms-25-05176-f004:**
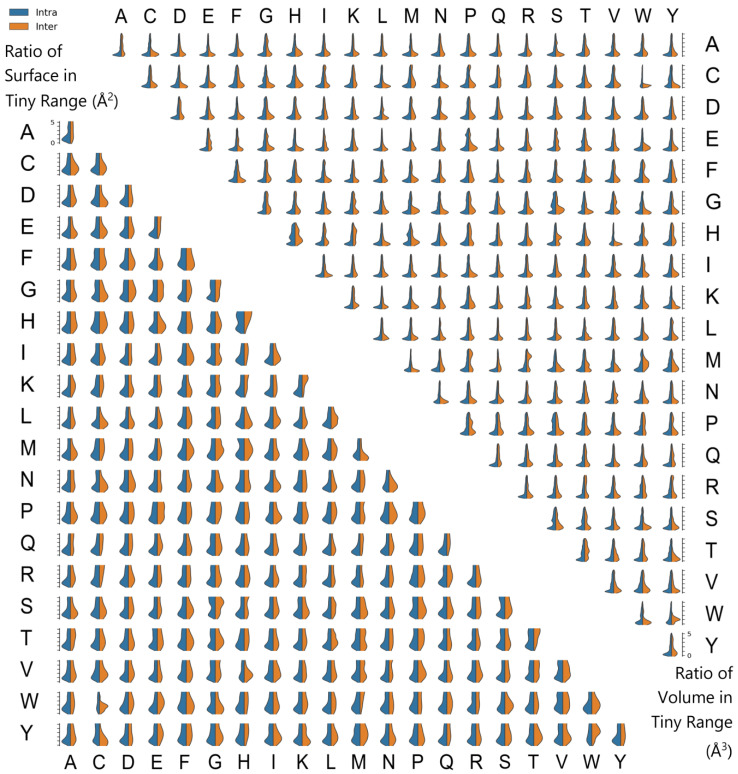
Distribution of small Contact Surface Areas and Volumes in NMR Protein Structures. Contact surface areas and volumes less than 5 Å^2^/Å^3^ were aggregated into bins with a size of 0.1 Å^2^/Å^3^ for percentage calculations (excluding the “-p” switch). The bin distributions for various types of inter-residue and intra-residue contacts are presented. The lower left corner illustrates the contact areas for inter- and intra-residue interactions in the range of 0 to 5 Å^2^. The upper right corner depicts the contact volumes for inter- and intra-residue interactions in the range of 0 to 5 Å^3^. ARIP allows for the customization of lower cutoffs to filter out interactions deemed invalid.

**Figure 5 ijms-25-05176-f005:**
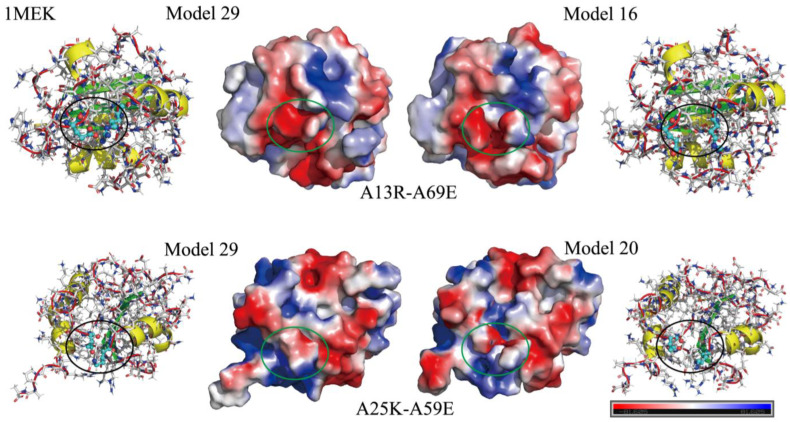
Charge distribution in Model 16 & 29 for A13R–A69E and in Model 20&29 for A25K–A59E of 1MEK. In Model 29, the two contacts A13R–A69E and A25K–A59E are very strong. However, in Model 16 & 20, these interactions are disrupted, leading to RMSD values of 1.52 Å and 1.34 Å for Model 16 & 29 and Model 20 & 29, respectively.

**Figure 6 ijms-25-05176-f006:**
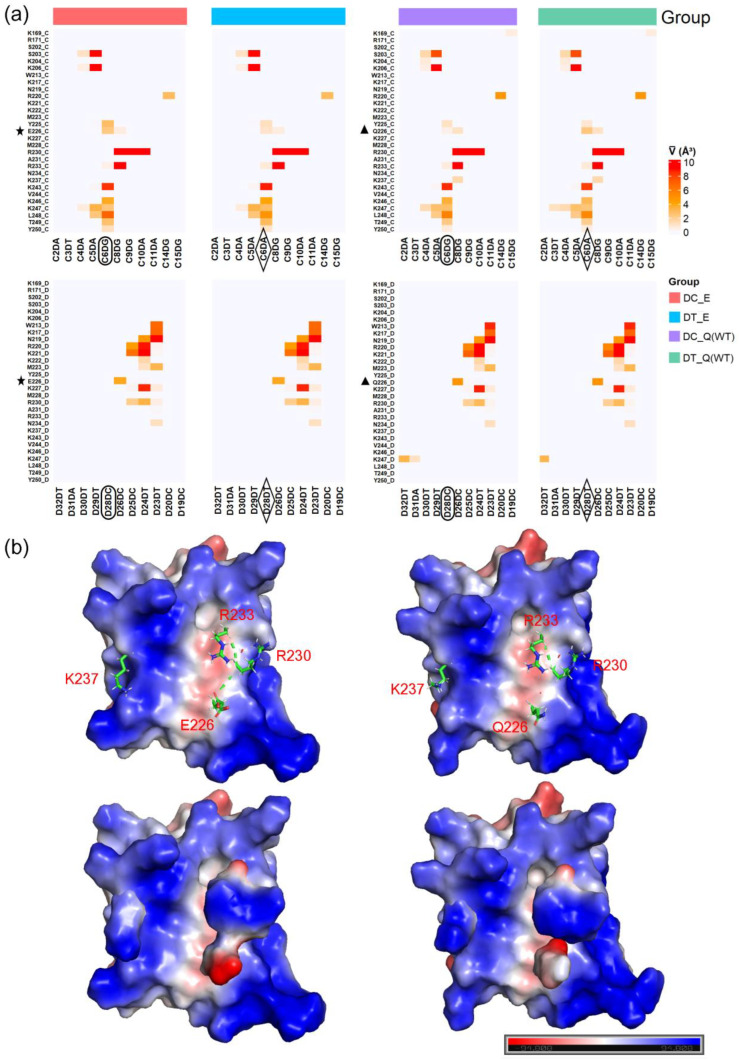
Protein SPI1 and DNA binding complexes. (**a**) Key interactions between protein SPI1 and DNA from the four groups, labeled with color bars. The key residues and basic groups for group classification are labeled with pentagrams, ovals, rhombi, and triangles. (**b**) Charge distribution models of SPI1 (Wild Type, PDB ID: 8E3K) and its Q226E mutation (PDB ID: 8EK3).

**Table 1 ijms-25-05176-t001:** Feature comparison between ARIP and dr_sasa.

	ARIP	dr_sasa
Supports proteins	Y	Y
Supports DNA/RNA	Y	Y
Guessing of unknown atom radii	Y	Y
Supports SASA calculation	Y	Y
Supports per atom analysis	Y	Y
Supports per atom detailed output	Y	Y
Supports per NMR model detailed output	Y	
Supports contact volumes calculation	Y	
Supports atomic overlap weighted algorithm	Y	
Supports hydrophilic atoms interactions mediated by water molecule algorithm	Y	
Supports dihedral angles calculation	Y	
Customization of lower cutoffs	Y	
Save as compressed format	Y	
Source code available	Y	Y
Offline usage	Y	Y

Y represents Yes.

**Table 2 ijms-25-05176-t002:** Average runtime of ARIP in different modes.

Input	Mode	Runtime
X-ray	Default	938.78 s
	Enhanced_precision	4810.20 s
	Surface	402.98 s
	Surface_enhanced_precision	960.35 s
NMR *	Default	360.14 s
	Enhanced_precision	1981.64 s
	Surface	154.85 s
	Surface_enhanced_precision	366.41 s

* NMR analysis time is calculated per model. The shorter time compared to X-ray may be due to the limitation of the NMR method in measuring proteins with smaller molecular weights.

**Table 3 ijms-25-05176-t003:** Distance range of residue interactions.

Location	Distance Range	Number of Separated Residues
Inner-chain	S	1~2
	M	3~4
	L1	5~10
	L2	11~20
	L3	21~30
	L4	31~40
	L5	41~50
	L6	>50
Inter-chain	I	Not on the same peptide chain

## Data Availability

A Python-based source code is freely available from https://github.com/YUYE-RainLeaf/ARIP/ (accessed on 6 May 2024).

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
