# Peer review of "ARIP: A Tool for Precise Interatomic Contact Area and Volume Calculation in Proteins"

_ijms, 2024, doi:10.3390/ijms25105176_

Round 1

Reviewer 1 Report

Comments and Suggestions for Authors

The paper describes a software calculating protein surfaces and volumes in a way that appears to be an enhanced version of other such methods. However, the present version is lacking significant clarity.

First of all, the authors never defined the ‘interaction surface’ and ‘interaction volume’. They refer to algorithms calculating sasa, i.e., solvent-accessible surface, but what does ‘interaction’ refer to? Interaction between two proteins? Or between different domains of the same protein?

Furthermore, most figures lack the definition of the axes, leaving the reader to guess.

I suggest that authors carefully revise the manuscript to make clear at the outset the definition of the properties the program calculates and make sure that all figures have their axes properly defined.

Author Response

Thank you very much for your thoughtful and constructive feedback on our manuscript, ijms-2951271. We greatly appreciate the time and effort you have dedicated to reviewing our work, and we are delighted to hear that the revisions have met your expectations.

The definitions of “interaction surface” and “interaction volume” have been provided in the second and third paragraphs of the revised version: “In these definitions, each atom is considered as a sphere with a radius equal to the sum of its own van der Waals radius and an additional distance, allowing for potential overlap between two spheres. The contact area refers to the surface area of one sphere that lies within another.” “The contact volume denotes the volume of one sphere encompassed by another, with the sphere serving as a visualization of an atom.” Based on these definitions, which are atom-based, ARIP can also derive interaction features for two proteins or between different domains of the same protein.

The axis definitions for all related figures have been added in the new version.

We are looking forward to the possibility of our manuscript being published and are grateful for the instrumental role your guidance has played in bringing it to this stage. Thank you very much!

Reviewer 2 Report

Comments and Suggestions for Authors

The manuscript presents a new analysis method. The method is quite interesting from a scientific standpoint and will bring innovations. However, what concerns me is the fact that they specified the method is compatible with nucleic acid and small molecule but did not provide any results to support this claim. From my perspective, the article can be published if the authors somehow demonstrate this compatibility.

 -          In the introduction, the authors wrote in lines 72 and 73, "...ARIP is comprehensive and user-friendly software that supports nucleic acids or other molecules stored in PDB format." However, in the results, the authors did not present any results showing this compatibility with nucleic acids and small molecules. The authors should add at least one example in the article for each of these that shows the compatibility.

-          The authors' idea is very good. Currently, the software is being made available at https://github.com/YUYE-RainLeaf/ARIP/. It would be interesting if the authors set up a web server for this in the future. It would increase the tool's usage not only for researchers knowledgeable in computational aspects.

-          In line 63, the authors mentioned that high-resolution (2Å) PDB structures were used. However, there are better resolutions deposited in the PDB; I suggest the authors replace the term "high resolution" with "good resolution."

-          The authors addressed the issue of stereo-hindrance in the results analysis. However, they did not clearly state the criteria considered for the stereo-hindrance effect. The authors should add this information somewhere in the article for clarification to readers.

-          In lines 111-113, the authors stated, “The ARIP algorithm ensures that a water molecule is assigned to mediate interactions between pairs of atoms with hydrophilic atoms within 2.8Å distance, maintain in the specified threshold.” Why was the distance chosen up to 2.8 Å when it is possible to mediate the interaction up to about 3.2 Å? What criteria was chosen? Add this information to the manuscript.

-          In lines 123-124, the authors claimed, “In our tests, dr_sasa often encountered errors and interrupted when analyzing NMR PDBs”. The authors should specify the errors encountered and clarify that the ARIP algorithm does not make these errors.

Comments on the Quality of English Language

Minor editing of English language required

Author Response

Thank you very much for your thoughtful and constructive feedback on our manuscript, ijms-2951271. We greatly appreciate the time and effort you have dedicated to reviewing our work, and we are delighted to hear that the revisions have met your expectations.

 In the revised version, we have analyzed multiple SPI1 structures from the study titled “DNA selection by the master transcription factor PU.1” as a case study for nucleic acid binding analysis. The interactions between the protein and DNA have been demonstrated. Our work revealed an enhanced residue-residue contact between R230 and R233 in SPI1 with the E226 mutation. Additionally, we found that K237 is critical for SPI1-DNA binding, as it loses its DNA-binding capability in the E226 mutation context.

Furthermore, CYP17A1 structures bound with drugs were used as a case for small molecule analysis. We observed that the interaction volume range (Max-Min) for N202 is significantly larger with galeterone (5.46 ų) compared to abiraterone (1.52 ų), suggesting a less stable interaction for galeterone.

 Indeed, we plan to establish a web server for ARIP in the future once we have a dedicated server for external services.

We have made the replacement in the modified version.

In the modified version, we have analyzed the interactions of water molecules with the atoms in the peptide plane to address the issue of steric hindrance introduced by these water molecules.

We have provided detailed explanations and descriptions of our methods in the modified version.

Reviewer 3 Report

Comments and Suggestions for Authors

Ma et al., present a revision of dr_sasa an algorithm for contact maps within proteins. I found the manuscript concise for the most part. Thus, instead of providing specific suggestions and commentary I'm going to develop on the main concern I have.

After careful reading of the manuscript I struggle to differentiate ARIP from others tools. This does not necessarily imply that it is not a substantial contribution. Perhaps with further details and in depth discussion this can be better achieved.

Similarly, the intended usage for ARIP is quite vague. Indeed, the study of protein-protein interactions and contacts is of high interest but the provided example fails to convey how exactly does ARIP excels over other tools. In this sense, I also ponder on the exact nature of the described volumes. Can these be expanded to include pharmacophoric information?

The authors also state that ARIP can be used to gain knowledge on protein function; then again they don't develop further on how can this be achieved with ARIP.

Perhaps the manuscript should be better focused on a case study. Plus, it is stated that ARIP is suited to study molecular dynamics trajectories. Thus, my suggestion is a comparison with the following reference: https://www.pnas.org/doi/10.1073/pnas.2017427118

I do think that the resulting work would be compelling and would provide a clear scope and capabilities of ARIP.

Comments on the Quality of English Language

There are some grammar and style errors in the manuscript. Please make a careful revision.

Author Response

Thank you very much for your thoughtful and constructive feedback on our manuscript, ijms-2951271. We greatly appreciate the time and effort you have dedicated to reviewing our work, and we are delighted to hear that the revisions have met your expectations.

I think that the interaction volume is a novel feature of ARIP. Through the analysis of structures determined by NMR technology, we have observed that the distribution curve for interaction volumes (Fig. 4) between amino acids is distinctly different from that of interaction areas. For each pair of residue-residue contacts, there is a single sharp peak near zero for the interaction volume, indicating that interaction volume may be a reliable factor for evaluating true physical contacts.

In the study, considering the vast array of conformations for each gene as described in the PNAS paper, we have chosen to focus on four specific states of the protein BRD4 as a representative case for analysis using molecular dynamics (MD) simulations. By analyzing the average and the range (Max-Min) of the interaction volumes, we have identified key residue-residue contacts (D106-T109, D106-P86, and M107-Q84) that are involved in conformational transitions, as reported in their paper. Additionally, another cluster of contacts (N121-E124, R58-W120, and R58-N117) located in a different loop has been detected, which may also play a significant role in conformational changes. Notably, R58, like D106, is sensitive to pH changes and could be a critical residue in mediating these transitions.

Round 2

Reviewer 1 Report

Comments and Suggestions for Authors

The paper continues to be lacking clear definitions of the concepts used/introduced. I will list several places where clarifications are needed, but I suggest the authors go through the manuscript with the mindset of a reader who was not involved in the development of these ideas.

L60: The contact area still is not defined. Also, it should be stated if the sphere radius is the VdW radius or the VdW radius incremented. If the latter, the increments should be specified – is it constant, or atom type dependent?

L61-62: I don’t see how the contact area is related to the Coulomb force.

L67: I don’t think ‘holistic’ is the right word – perhaps ‘complete’ would be appropriate here.

L116: Where does the term ‘mosaic distance come from? If it is a known concept, a reference is needed. If it is introduced here than it has to be introduced in a clear way, not as a half sentence that is not even clear or worse. According to the figure, when it is positive, it is the shortest distance between two non-overlapping spheres; when it is negative then if is the longest distance between the two overlapping section of the sphere surfaces. Furthermore, what are the radii (see above)? Is the water radius 1.4 A?

L120: So now ‘mosaic distance is shortened to distance? Please, clarify.

L194: What is the meaning of ‘integrated for comparison’? Can’t you just say ‘analyzed’?

Figure 3 needs a lot of clarifications. (1) There are three color sets – which goes where? Also, most reader would not be able to clearly distinguish that many colors – I certainly can’t. (2) What is the horizontal axis? (3) What are the bars on the top of the figure?

Figure 4 needs a lot of clarifications. (1) What is the ‘tiny range’? (2) What do those funny shaped figures represent? I understand that the colors represent two different distributions but I don’t know what? (3) What is ‘Ratio of volume’? Ratio to what?

L:300: What are ‘SPI1-DNA interactions’? Also, why use color bars on top of the tables instead of the labels the colors represent?

In Eq(1) I the sum over 1/ci needs some explanation/justification. If we take the simples case of two atoms then ci is uniformly to, thus the formula would give the sum of the grid volumes in the intersection divided by two. Why?

Discussion: It seems to me that this tool can also be used to check the goodness of a structure, be it experimental or computational: If too many calculated contact parameters fall to the fringes of their distribution then the structure can be called problematic.

As for the waters, it could be mentioned that not all waters are ‘seen’ by x-rays.

Author Response

We are sincerely grateful for your comprehensive and insightful feedback on our manuscript, ijms-2951271. Your thorough review and the attention you have given to our work are greatly appreciated. We are thrilled to hear that our revisions have met your expectations.

L60: The contact area still is not defined. Also, it should be stated if the sphere radius is the VdW radius or the VdW radius incremented. If the latter, the increments should be specified – is it constant, or atom type dependent?

Reply: Thank you for your suggestion. The definition of “contact area” is indeed addressed in the third paragraph, where it is defined as “the surface area of one sphere that lies within another.” It should be noted that if two contacting atoms are of different types, their van der Waals (VdW) radii will differ, leading to distinct contact areas for each atom. In the context of ARIP, the parameter “-p” is essential; when included, it applies a fixed solvent radius of 1.4 Å exclusively to hydrophilic atoms such as oxygen (“O”), nitrogen (“N”), and sulfur (“S”). For default settings, all atoms, including hydrophilic carbon atoms, are assigned a fixed solvent radius of 1.4 Å. It is important to clarify that the sphere radius is equal to the sum of its own VdW radius and the fixed solvent radius when applicable.

This response offers a clear explanation of the contact area definition and the parameters involved in its calculation within the ARIP tool. It also addresses the need for specificity regarding the use of VdW radii and any additional solvent radius applied.

L61-62: I don’t see how the contact area is related to the Coulomb force.

Reply: Thank you for your suggestion. We appreciate the reviewer's inquiry regarding the relationship between contact area and Coulomb force. In our discussion, we propose a hypothetical link between these two properties based on physical principles. The Coulomb force, which governs electrostatic interactions between charged particles, varies inversely with the square of the distance between them. Similarly, the surface area of an object is directly proportional to the square of its radius. While these are not direct analogs, we suggest that the contact area, as a measure of the interface between two atoms, could be indicative of the potential for electrostatic interactions, albeit in a more complex biological context.

Moreover, we introduce the concept of contact volume to provide an additional dimension to this analysis. Unlike contact area, which is a two-dimensional measure, contact volume inherently includes the depth of the interaction, which could be more closely related to the energetics of the interaction. This is because the volume potentially represents the space available for electron movement, which could be associated with the binding energy between the atoms.

It is important to emphasize that these are preliminary assumptions and the actual calculation of forces and energies between atoms is indeed complex. The interactions are influenced by various factors, including the chemical environment, the presence of other atoms or molecules, and the specific properties of the atoms in question. Our use of contact area and volume is an attempt to introduce parameters that could be useful in the quantitative analysis of interatomic interactions in proteins, with the understanding that further research and validation are required to fully understand their biological significance.

We acknowledge that the relationship between contact area, Coulomb force, and contact volume with energy is not straightforward and may not be linear. The purpose of our study is to initiate a discussion on these parameters and to invite further exploration into their potential applications in understanding protein structure, function, and interactions. We believe that the inclusion of these parameters in our analysis with ARIP provides a foundation for future research that could lead to a more nuanced understanding of molecular interactions.

L67: I don’t think ‘holistic’ is the right word – perhaps ‘complete’ would be appropriate here.

Reply: Thank you for your suggestion. We have made the replacement in the modified version.

L116: Where does the term ‘mosaic distance come from? If it is a known concept, a reference is needed. If it is introduced here than it has to be introduced in a clear way, not as a half sentence that is not even clear or worse. According to the figure, when it is positive, it is the shortest distance between two non-overlapping spheres; when it is negative then if is the longest distance between the two overlapping section of the sphere surfaces. Furthermore, what are the radii (see above)? Is the water radius 1.4 A?

Reply: We thank the reviewer for their insightful comments and acknowledge the need for clarity regarding the term 'mosaic distance.' This concept is indeed introduced within the context of our study and is used to describe the spatial overlap between two atomic van der Waals (VdW) radii. As depicted in Figure 2f, the mosaic distance represents the extent of overlap between two spheres, which can be considered as the negative distance when they are superimposed.

In the provided figure, when the mosaic distance is positive, it refers to the shortest distance between the surfaces of two non-overlapping spheres. Conversely, when the mosaic distance is negative, it indicates the longest distance within the region where the two spheres' surfaces overlap. This concept is central to our analysis of interatomic interactions and is detailed further in the section labeled L116 of our manuscript.

Regarding the radii, we clarify that the term 'mosaic distance' specifically refers to the overlap between atomic radii. For water molecules, we adopt the radius of the oxygen atom, which is 1.4 Å, as a proxy for the water molecule's radius in our calculations. This approximation is based on the recognition that the hydrogen atoms in water are significantly smaller and can be approximated by the radius of the oxygen atom, which is the dominant contributor to the water molecule's size.

We appreciate the opportunity to refine our explanation and will ensure that the term 'mosaic distance' is introduced clearly and concisely in the manuscript to avoid any ambiguity. We will also provide a more thorough explanation of the radii used in our calculations, ensuring that the reader can easily understand the basis of our methodology and its application in the context of our study.

In summary, the term 'mosaic distance' is introduced to describe the overlap between atomic spheres, and the radii used in our calculations are based on established atomic and molecular sizes, with specific mention of the 1.4 Å radius for water molecules. We have revised the manuscript to present this information more clearly and to ensure that the reader can easily understand the basis of our methodology and its application in the context of our study.

L120: So now ‘mosaic distance is shortened to distance? Please, clarify.

Reply: Thank you for your clarification, reviewer. In light of your feedback, we have made the following adjustments in the updated version of our manuscript to distinguish between the two types of distances we are examining:

'Spherical distance' is now used to denote the distance measured between the surfaces of two atomic spheres, which reflect the extent of overlap between the spheres when they are in contact.

'Center distance' refers to the distance between the centers of the two atomic spheres, which is a measure of the separation between the geometric centers of the atoms in question. We have refined our use of 'mosaic distance' to 'distance' in the context of atomic contact to avoid confusion. This term now specifically refers to the overlap distance between two atomic spheres as they interact, which is a critical parameter in our analysis of interatomic interactions. The manuscript has been revised to include clear definitions and to use these terms consistently. We believe these changes will provide the necessary clarity and precision to the reader regarding the measurements and calculations involved in our study. We are grateful for the opportunity to enhance the quality and understanding of our research through your feedback.

L194: What is the meaning of ‘integrated for comparison’? Can’t you just say ‘analyzed’?

Reply: Thank you for your suggestion. We have rewritten that portion of result in the modified version.

Figure 3 needs a lot of clarifications. (1) There are three color sets – which goes where? Also, most reader would not be able to clearly distinguish that many colors – I certainly can’t. (2) What is the horizontal axis? (3) What are the bars on the top of the figure?

Reply: We appreciate the reviewer's feedback on Figure 3 and acknowledge the need for further clarification to ensure the figure is accessible and comprehensible to all readers. In response to the concerns raised, we have implemented the following revisions and clarifications in the updated version of our manuscript:

Color Coding Clarification: We have revised the color scheme to simplify the representation and improve readability. Initially, we used yellow to represent the value zero, but we have now changed this to white for better clarity. The color blue is used to indicate smaller percentages, while red signifies larger percentages. This color coding is intended to highlight the distribution of peak values within the figure. We have included a key within the figure to assist readers in distinguishing the colors and their corresponding values.

Horizontal Axis Description: The horizontal axis represents specific types of atoms within a certain amino acid. It is designed to provide a detailed view of the atomic interactions, complementing the broader statistical distribution presented in Figure 2. Each bar on the horizontal axis corresponds to a specific atom type within an amino acid residue, as detailed in the Methods section.

We thank the reviewer for their valuable suggestions, which have helped us to improve the presentation of our research.

Figure 4 needs a lot of clarifications. (1) What is the ‘tiny range’? (2) What do those funny shaped figures represent? I understand that the colors represent two different distributions but I don’t know what? (3) What is ‘Ratio of volume’? Ratio to what?

Reply: We appreciate the reviewer's feedback and recognize the need for further clarification regarding Figure 4. Here are the detailed explanations in response to the specific points raised:

(1)'Tiny Range' Explanation: The 'tiny range' refers to the specific and narrow range of interaction volumes and areas that we have chosen to focus on for detailed analysis. This range, from 0 to 5 Ų/ų, is selected to identify and characterize the most biologically relevant interactions within the protein structure. By focusing on this range, we aim to distinguish the true interactions from the background noise, essentially establishing a threshold for interaction significance.

(2)Representation of Subfigures: Each subfigure within Figure 4 represents a distinct distribution of interaction volumes and areas. These distributions are derived from the analysis of the specified 'tiny range' and are intended to provide insights into the frequency and significance of different interaction types within the protein structure. The colors used in the figure differentiate between two types of interactions: inter-chain (between different polypeptide chains) and intra-chain (within the same polypeptide chain).

(3)'Ratio of Volume' Definition: The term 'Ratio of volume' is part of the 'Ratio of volume In Tiny Range' analysis. It represents the proportion of interaction volumes within the 'tiny range' of 0 to 5 ų relative to the total number of interactions considered in the study. This ratio is calculated by binning the interactions into increments of 0.1 ų and determining the count of interactions within each bin. The ratio for each bin is then computed as the number of interactions in that bin divided by the total count of all interactions, yielding a measure of the relative frequency of interactions within that specific volume range.

Color Coding and Bin Analysis: The two colors in the figure are used to distinguish between the two types of interactions mentioned above. Each color, therefore, represents a different distribution of interaction volumes and areas within the 'tiny range'.

We have revised the manuscript to include these clarifications and have updated the figure legend and the methods section to ensure that the 'tiny range', the representation of subfigures, and the calculation and meaning of the 'Ratio of volume' are explicitly defined and easily understandable. Our goal is to present the data in a clear and accessible manner, and we are grateful for the opportunity to enhance the clarity of our presentation.

L:300: What are ‘SPI1-DNA interactions’? Also, why use color bars on top of the tables instead of the labels the colors represent?

Reply: We appreciate the reviewer's inquiry regarding the representation of 'SPI1-DNA interactions' and the choice of color bars in our figures. Here are the clarifications:

SPI1-DNA Interactions: SPI1, also known as PU.1, is a transcription factor that plays a crucial role in immune cell development and is known to interact with DNA. The term 'SPI1-DNA interactions' specifically refers to the molecular interactions between the SPI1 protein and the DNA sequences it binds to regulate gene expression. These interactions are essential for the function of SPI1 in cellular processes.

The color bars above the tables are utilized to provide a visual reference that corresponds to the data presented within the tables. The use of color bars serves several purposes:

Enhanced Differentiation: Colors can be more easily distinguished than text labels, especially when readers are quickly scanning or comparing data across multiple rows or columns.

Visual Clarity: Color bars offer a clear and immediate association between the data and their corresponding categories or values, reducing the potential for confusion or error.

Legend and Description: We have included a comprehensive legend and description in the methods section to explain the meaning of each color bar and how it relates to the data in the tables. This ensures that readers can understand the significance of the color coding in the context of the study.

In Eq(1) I the sum over 1/ci needs some explanation/justification. If we take the simples case of two atoms then ci is uniformly to, thus the formula would give the sum of the grid volumes in the intersection divided by two. Why?

Reply: We appreciate the reviewer’s feedback. In fact, ci represents the count of atoms in contact with a given atom within a grid. If ci equals 2, it means there are three atoms in contact in the target grid. This method is central to the dr_sasa algorithm and enhances the accuracy of contact surface area (CSA) calculations. For further clarification, Figure S3 in the dr_sasa paper illustrates the algorithm for calculating CSA in a three-body system. The red, blue, and purple regions represent parts of the surface of body 1 that are buried by bodies 2 and 3, respectively. Specifically, the purple region indicates surface points that are buried by bodies 2 and 3, resulting in ci equaling 2, which prevents double counting in the calculation of the buried surface area (BSA).

Discussion: It seems to me that this tool can also be used to check the goodness of a structure, be it experimental or computational: If too many calculated contact parameters fall to the fringes of their distribution then the structure can be called problematic.

Reply: We are grateful to the reviewer for their insightful suggestion regarding the potential application of our tool in assessing the quality of protein structures. The tool’s capability to calculate contact parameters indeed provides valuable insights into the reliability of both experimental and computationally derived protein structures.

We appreciate the reviewer’s constructive feedback and will consider the potential of our tool for evaluating protein structures in our ongoing research. This suggestion opens up an exciting avenue for the application of our tool and could have significant implications for the validation and refinement of protein structures.

As for the waters, it could be mentioned that not all waters are ‘seen’ by x-rays.

Reply: We acknowledge the reviewer's comment regarding the visibility of water molecules in X-ray crystallography. Indeed, not all water molecules are discernible in every crystal structure, which can impact the accuracy of solvent radius measurements and subsequent structural analyses. In our study, we addressed this limitation by employing a statistical approach.

To elucidate the behavior of water molecules around protein atoms, we analyzed a large dataset comprising several thousand protein and nucleic acid structures with water molecules resolved by X-ray crystallography. By examining the collective distribution patterns, we aimed to capture the general trends and properties of water molecules interacting with protein atoms. This methodological choice allowed us to account for the variability in water molecule visibility and to minimize the impact of noise in our analysis.

During this process, we calculated ratios to normalize the data and to eliminate inconsistencies that may arise from the inability to visualize every water molecule. This statistical treatment of the data provides a robust estimate of the average positions and interactions of water molecules, which are crucial for understanding the solvation characteristics and the dynamic behavior of proteins.

We have clarified this aspect in the revised manuscript to ensure that the reader understands the rationale behind our approach and the measures taken to ensure the reliability of our findings. We believe that this approach provides a comprehensive and accurate representation of the interactions between water molecules and protein atoms, contributing to a better understanding of protein structure and dynamics.

We appreciate the reviewer's feedback and have made the necessary revisions to the manuscript to reflect the considerations regarding the visualization of water molecules in X-ray crystallography.

Reviewer 2 Report

Comments and Suggestions for Authors

The authors made the changes. The manuscript can be accepted in current version.

Author Response

Thank you very much!

Reviewer 3 Report

Comments and Suggestions for Authors

I thank the authors for acknowledging my suggestions. While I do think the work has been improved, noticeable problems persist.

A recurrent issue that I found was the lack of details or proper explanation of things. As an example, I will focus on the added text regarding BRD4:

The authors talk about BRD4 and the previous study. But in the following lines information is presented in quick succession .

As a reader I would be baffled. By the end of the paragraph I just learned that BRD4 is related to histones and that ARIP found the same contacts from the previous study, plus some novel ones.

Which holds merit on its own right; then again, as a reader all I gained was a little factoid. The text fails to convey why should this hold any significance.

Why is BRD4 important?

What does ARIP bring to the discussion when compared to the methods of

Raich et al?

Authors may argue that for  this last question  a straight answer is given. Yet, as a counter argument I must say that the brief mention of additional contacts is not provided within the context of BRD4 and a rather vague conjecture referencing pH. Such would be the difference between a mere example and a true case study.

This can be extrapolated to the manuscript. The work is interesting yes, but the text fails to convey its true significance on the big picture. Please critically assess an extensive revision and revap of the main text.

Comments on the Quality of English Language

Some errors persist within the text. Please revise.

Author Response

We are sincerely grateful for your thoughtful and constructive feedback on our manuscript, ijms-2951271. Your dedication of time and effort in reviewing our work is greatly appreciated, and we are pleased to hear that the revisions have met your expectations.

In our revised analysis, we have incorporated essential background information to contextualize our findings for each protein of interest. For BRD4, we have identified residue R58, analogous to D106, as a potential pH-sensitive site. This inference is grounded in the understanding that BRD4 operates within the cell nucleus, an environment where the juxtaposition of alkaline histones and acidic DNA suggests that fluctuations in environmental pH could significantly influence BRD4's conformation and, by extension, its functional interactions.

To substantiate our observations on the pH dependence of PDI, we conducted a thorough review of the literature, examining several published papers to validate our findings. This comprehensive approach ensures that our conclusions are not only well-supported by experimental data but also aligned with the current scientific consensus.

Round 3

Reviewer 1 Report

Comments and Suggestions for Authors

The authors addressed all of my concerns in the revised manuscript.